# A Label-Free Fluorescent DNA Machine for Sensitive Cyclic Amplification Detection of ATP

**DOI:** 10.3390/ma11122408

**Published:** 2018-11-29

**Authors:** Jingjing Zhang, Jialun Han, Shehong Feng, Chaoqun Niu, Chen Liu, Jie Du, Yong Chen

**Affiliations:** 1State Key Laboratory of Marine Resource Utilization in South China Sea, College of Information Science & Technology, Hainan University, Haikou 570228, China; zhangjingjingaoxue@163.com (J.Z.); jialun_han@126.com (J.H.); Niucq@foxmail.com (C.N.); liuchen8642@163.com (C.L.); 2College of Materials & Chemistry Engineering, Hainan University, Haikou 570228, China; 3Institute of Tropical Agriculture and Forestry, Hainan University, Haikou 570228, China; 990147@hainu.edu.cn

**Keywords:** DNA machine, ATP detection, label-free fluorescence, cyclic amplification, graphene oxide, logic gate

## Abstract

In this study, a target recycled amplification, background signal suppression, label-free fluorescent, enzyme-free deoxyribonucleic acid (DNA) machine was developed for the detection of adenosine triphosphate (ATP) in human urine. ATP and DNA fuel strands (FS) were found to trigger the operation of the DNA machine and lead to the cyclic multiplexing of ATP and the release of single stranded (SS) DNA. Double-stranded DNA (dsDNA) was formed on graphene oxide (GO) from the combination of SS DNA and complementary strands (CS′). These double strands then detached from the surface of the GO and in the process interacted with PicoGreen dye resulting in amplifying fluorescence intensity. The results revealed that the detection range of the DNA machine is from 100 to 600 nM (R^2^ = 0.99108) with a limit of detection (LOD) of 127.9 pM. A DNA machine circuit and AND-NOT-AND-OR logic gates were successfully constructed, and the strategy was used to detect ATP in human urine. With the advantage of target recycling amplification and GO suppressing background signal without fluorescent label and enzyme, this developed strategy has great potential for sensitive detection of different proteins and small molecules.

## 1. Introduction

Adenosine triphosphate (ATP) plays an important role in various cellular metabolism processes and biochemical reactions in cells and biological organisms. Many diseases are related to ATP concentration, such as ischemia, hypoxia, Parkinson’s disease, hypoglycemia, and some malignant tumors [1,2,3]. Therefore, the study of a strategy for ATP detection is of clinical significance. The use of nucleic acid aptamers has unique advantages in clinical diagnosis and sensing applications because they are low cost, easy to synthesize, and have unique characteristics, such as good chemical stability and long storage times [4,5], and for these reasons, they have been widely studied [6,7].

In recent years, signal amplification methods, such as single primer isothermal amplification [8], rolling circle replication [9], DNA enzyme [10,11], artificial electromagnetic metamaterials and near-zero-index materials [12,13], and strand displacement amplification (SDA) [14,15,16,17], have been used to effectively improve the sensitivity of biosensors. However, most of these signal amplification strategies are usually influenced by large background signals and require the use of biological enzymes. Enzymes are often costly and are susceptible to the effects of temperature and pH. Quenchers, such as graphene oxide (GO) [18,19], magnetic nanoparticles [20], gold nanoparticles [21,22], metal-organic frameworks [23], and nanoceria [24], have been shown to inhibit background signals and improve sensor sensitivity. However, these methods require the use of fluorescent labeling and are also very expensive. Fluorescent dye probes, such as PicoGreen [25,26], thiazole orange (TO) [27], Thioflavin T [28,29], and SYBR Green I [30], have proved to be effective label-free fluorescence methods. Moreover, these label-free fluorescence methods do not usually use a signal amplification strategy or background signal suppression strategy to enhance the sensitivity of biosensors. In summary, it is difficult to achieve simultaneous signal amplification, background signal suppression, label-free fluorescence, and enzyme-free ATP detection. In this work, target ATP cyclic amplification, GO suppression background signals, and PicoGreen dye were used to construct a label-free fluorescent and enzyme-free DNA machine for the sensitive detection of ATP.

It is also worth noting that much of the research on molecular logic gates has focused on DNA, due to it having a well-regulated structure, straightforward hybridization rules, and molecular recognition abilities, and DNA machines have calculated some math problems [31,32]. Over the past decade, a series of molecular logic gates, including INH-NINH, AND, OR, and XOR [33,34], have been studied. The design of complex and integrated molecular logic gates has many potential advantages, especially when based on molecular systems such as DNA. In this work, using ATP, DNA fuel strands (FS), and complementary strands (CS′) as input signals and PicoGreen dye fluorescence intensity as an output signal, AND-NOT-AND-OR logic gates and a DNA machine circuit were constructed.

## 2. Experimental

### 2.1. Materials and Reagents

Uridine triphosphate (UTP), ATP, cytidine triphosphate (CTP), guanosine triphosphate (GTP), and PicoGreen dye were purchased from Shanghai Yi Sheng Biotechnology Co., Ltd. (Shanghai, China). Nucleotide sequences (Table 1) were purchased from the Beijing Genomics Institute (Beijing, China). A graphene oxide solution with a graphene oxide content of 1 wt.%, was supplied by Shanghai Aladdin Biochemical Technology Co., Ltd. (Shanghai, China).

All reagent dilutions were carried out with a buffer (10 mmol/L Tris, 50 mmol/L NaCl, 10 mmol/L MgCl_2_, pH 7.5). One micrometer of aptamer DNA strands (AS), 1 µM of SS, and 1 µM of helper DANS strands (HS) were mixed. The resulting mixture was subsequently heated to 95 °C for 5 min and then cooled down slowly to room temperature inside a furnace. Fuel stranded (FS) and complementary stranded (CS′) DNA were heated to 95 °C for 5 min and were then quickly placed into a refrigerator set to a temperature of around −20 °C for 5 min, respectively. PicoGreen was diluted 200 times and GO (1 wt.%) was diluted 100 times. In all of the experiments, the volume of the buffer was 1800 µL; the AS, SS, and HS mixture totaled 70 µL; the PicoGreen volume was 25 µL; and the GO volume used was 10 µL. The amount of FS and CS′ used was 40 µL in each case.

### 2.2. Apparatus

Fluorescence spectra and reaction times were measured using a fluorescence spectrometer (Model: RF-6000, Shimadzu, Osaka, Japan). The parameters used in the reaction time experiments were a 480 nm wavelength emission and excitation at 520 nm. The spectral parameters used were a 480 nm wavelength emission.

### 2.3. ATP Detection in Real Urine Samples

Urine samples were obtained from a healthy adult at Hainan University hospital (Haikou, China). The urine samples were centrifuged at 13,000 r/min for 3 min. The supernatants of the resulting samples were then diluted 10 times before ATP was incorporated into the diluted urine samples at concentrations of 100, 300, and 500 nM, respectively. The measurements were repeated three times to compute the urine recovery.

## 3. Discussion and Analysis

### 3.1. Construction of a DNA Machine

Scheme 1 shows a label-free fluorescent and enzyme-free DNA machine for the target recycled amplification background suppression detection of ATP. The sensing system involves the use of GO, PicoGreen dye, a DNA probe, and CS′ (the black sequence) and FS (the blue sequence) DNA. The DNA probe consists of AS DNA (the violet sequence), SS DNA (the red sequence), and HS DNA (the green sequence). In the absence of ATP, the fluorescence intensity was very weak due to the toehold of the DNA probe being adsorbed onto the surface of GO via π-π stacking and the generation of fluorescence resonance energy transfer (FRET) [18,19]. In the presence of the ATP target, the ATP binds to the AS DNA, releasing SS DNA, which then combines with CS′ DNA to form dsDNA (SS + CS′) strands that due to being weakly bound to GO, shed away from its surface. These double (SS + CS′) strands then interact with PicoGreen dye to produce a significantly amplified fluorescence signal [21]. Then, FS DNA crosses with AS DNA starting from the toehold of the DNA probe and simultaneously releases HS DNA and ATP. ATP again combines with the DNA probe to trigger the operation of the DNA machine, resulting in the cyclic multiplexing of ATP and the release of many single DNA strands [35].

### 3.2. The Feasibility of the DNA Machine

To demonstrate whether it is feasible to use the DNA machine for ATP detection, the fluorescence spectra of different mixed solutions were tested after being incubated for 3 h. Figure 1 shows that the lowest fluorescence intensity (a-curve) was observed due to the DNA probe being adsorbed onto the GO surface, resulting in FRET. Negligible fluorescence intensity changes (b-curve, c-curve vs. a-curve) were observed due to a lack of CS′ DNA and no fully complementary double strands. The fluorescence intensity change of the d-curve was also small because of a lack of target ATP and unreleased single strands of DNA. In the presence of the target ATP and CS′ strands, there was an obvious increase in the fluorescence intensity (e-curve vs. b-curve), indicating that ATP combined with the probe, releasing SS DNA to combine with CS′ DNA to form dsDNA (SS + CS′), which was then shed from the surface of the GO, and interacted with PicoGreen dye to produce a significantly amplified fluorescence signal. When ATP, FS DNA, and CS′ DNA were present at the same time, the changes in the fluorescence intensity (f-curve vs. e-curve) were more significant. Due to the presence of ATP and FS DNA, the DNA machine began to work and released more SS DNA to combine with CS′ DNA to form dsDNA, which was then shed away from the GO surface and interacted with PicoGreen dye to generate more obvious fluorescence intensity. The results show that the DNA machine is thus feasible for use in ATP detection.

In order to further prove the feasibility of the principle, the time detection of the reaction process was undertaken. According to the a-segment shown in Figure 2A, the fluorescence intensity remained at the weakest, which is the same as the result of the b-curve shown in Figure 1. According to the b-curve shown in Figure 2A, when an excess of CS′ DNA was added, there was an obvious increase in the fluorescence intensity. This proves that ATP binds to AS DNA and releases SS DNA, which also verifies the e-curve results shown in Figure 1. According to the c-curve shown in Figure 2A, when FS DNA was added, there was a further increase in the fluorescence intensity. This demonstrates that ATP and FS DNA trigger the operation of the DNA machine to release more SS DNA, which further validates the f-curve results shown in Figure 1. In the a-segment in Figure 2B, the fluorescence intensity was very low due to a lack of ATP and SS DNA release, consistent with the a-curve data, as shown in Figure 1. Subsequently, upon the addition of CS′ DNA, as observed from the b- and c-curves in Figure 2B, there was no significant change in the fluorescence intensity. This further confirms that the operation of the DNA machine requires ATP, which validates the d-curve results shown in Figure 1. In summary, the results shown in Figure 2 are consistent with those of Figure 1, which confirms that the principle of the DNA machine is feasible.

### 3.3. The Logic Gates and DNA Machine Circuit

In terms of the logic gates, the input signal is 1 if ATP, CS′ DNA, or FS DNA are present, if not, it is 0. Meanwhile, at a wavelength of 520 nm, if the change in the fluorescence intensity, ΔF, is greater than 45, the output signal is 1, if not, it is 0. Using the results of Figure 1, Figure 3A was compiled, which shows that in the presence of both ATP and CS′ DNA, there is an obvious change in the fluorescence intensity of the input signal (1,1,0). However, in the presence of ATP, CS′ DNA, and FS DNA there is a more obvious change in the fluorescence intensity of the input signal (1,1,1). A schematic diagram of the equivalent circuit of the DNA machine is shown in Figure 3B.

According to Figure 2B, it can be deduced that in the presence of CS′ DNA (0,1,0) or FS DNA (0,0,1) there is little change in the fluorescence intensity, and the output signal is 0. At the same time, using the results shown in Figure 3A, a truth table was compiled and is shown as Table 2. According to the digital logic, circuit and truth table information, the following can be deduced:
Out1(△F) = ATP·CS′·FS(1)
(2)$$\mathrm{Out}2(\Delta F)\;=\;\mathrm{ATP}\cdot \mathrm{CS}\prime \;=\;\mathrm{ATP}\cdot \mathrm{CS}\prime \cdot (\mathrm{FS}\;+\;\bar{\mathrm{FS}})\;=\;\mathrm{ATP}\cdot \mathrm{CS}\prime \cdot \mathrm{FS}\;+\;\mathrm{ATP}\cdot \mathrm{CS}\prime \cdot \bar{\mathrm{FS}}$$

According to Equations (1) and (2), the logic gates of the DNA machine can be represented in schematic form, as shown in Figure 4.

### 3.4. The Linearity of the DNA Machine

After the sensing solutions were incubated for 3 h, the fluorescence intensities of different ATP concentrations were tested using the described method. Figure 5A shows that an increase in the ATP concentration from 0 to 3000 nm led to a gradual increase in the fluorescence intensity.

According to Figure 5A, the relative change rate values, ΔF/F_0_, where F_0_ is the peak of the blank curve in Figure 5A, and ΔF is the peak difference between the other curves and the blank curve, were obtained, and are shown in Figure 5B. From Figure 5B, it can be observed that the data of the ATP samples with concentrations from 100 to 600 nM show better linearity. Therefore, the linear fitting of the 100 to 600 nM ATP concentrations was taken as the sensor linear detection range, as shown in Figure 5C. The fitting linear equation was Y = 6.86802 × 10 − 4X − 0.01174 (R^2^ = 0.99108) with a LOD of 127.9 pM (3σ/slope). Compared to most other methods (see Table 3), the results clearly indicate that the proposed method has a lower LOD, increased sensitivity and sufficient linearity for ATP detection.

### 3.5. The Selectivity of the DNA Machine

To examine the selectivity of the DNA machine, other analogous molecules (GTP, UTP, CTP, and a mixture of ATP, GTP, UTP, and CTP) were tested against ATP. As shown in Figure 6, when comparing ATP, CTP, UTP and GTP, there was a significant increase in the ΔF/F_0_ values of the ATP and the mixture, indicating that the DNA machine is highly selective for the detection of ATP.

### 3.6. Using the DNA Machine for ATP Concentration Detection in Real Urine Samples

In order to evaluate the application of the DNA machine for ATP detection in real urine, recovery tests were carried out. The urine of healthy people was centrifuged, and the supernatant was diluted 10 times using a buffer solution. Different ATP concentrations were added to the dilute liquid and tested using the described method. Table 4 shows that the DNA machine exhibits good recoveries in the range of 95.76–102.03%, indicating that the proposed DNA machine has great potential for ATP detection in real urine samples.

## 4. Conclusions

In summary, a target recycled amplification, background signal suppression, label-free fluorescent, enzyme-free DNA machine was developed for highly sensitive ATP detection. The DNA machine is triggered in the presence of both the target ATP and FS DNA. Target ATP cycle amplification and GO background signal suppression strategies are used simultaneously to improve the sensitivity of the DNA machine. The detection range was found to be from 100 to 600 nM (R^2^ = 0.99108) with a LOD of 127.9 pM. This method has strong selectivity and is very suitable for ATP detection in real human urine. On the basis of this strategy, AND-NOT-AND-OR logic gates and a DNA machine circuit were successfully constructed. This method has the characteristics of being low cost due to not requiring any fluorescent labeling and biological enzymes. This recommended method could also be adopted for highly sensitive detection of various ions and proteins. In the future, the sensitivity of the DNA machine for ATP detection can be improved by catalyzing hairpin assembly.

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
