# Peer review of "A Label-Free Fluorescent DNA Machine for Sensitive Cyclic Amplification Detection of ATP"

_materials, 2018, doi:10.3390/ma11122408_

Round 1
Reviewer 1 Report
This paper is suitable for publication on this international Journal but after minor revisions listed below:
Please, Could the authors add more details about the DNA machine's characterization, under a material science point of view.
The second issue is: a comparative study with the conventional analytical systems, widely applied for the ATP detection.
Another aspect concerns the English language presentation. For me, it will be require a mother tongue revision, especially for the sentence constructions, idiomatic expression and a more synthetic style, get to the point expeciilly for conclusions, extracted from results and experimental data.
Author Response
Dear Professor/editor,
Thank you very much for your attention to our article “A label-free fluorescent DNA machine for sensitive cyclic amplification detection of ATP” (Manuscript ID: materials-394170). The comments from referees are very valuable and helpful for improving our article. All the authors have seriously discussed about these comments. We have accordingly revised the manuscript to meet with the requirements of the journal “Materials”. Our changes are included in the revised manuscript marked by the RED font and the detailed point-by-point responses to the comments are listed as RED font as following.
To Referee: 1
This paper is suitable for publication on this international Journal but after minor revisions listed below:
Please, Could the authors add more details about the DNA machine's characterization, under a material science point of view.
Thank you for your positive comments. The DNA machine was a liquid system including AS, HS, CS', SS, GO and PicoGreen dye, not solid powder. So, it is difficult to be studied by SEM, XPS, etc.
The second issue is: a comparative study with the conventional analytical systems, widely applied for the ATP detection.
Thank you for your valuable suggestions to improve the quality of our manuscript. In the conventional analytical systems, it is difficult to achieve the simultaneous signal amplification, background signal suppression, label-free fluorescence, enzyme-free ATP detection. In this work, target ATP cyclic amplification, GO suppression background signals, and PicoGreen dye were used to construct a label-free fluorescent and enzyme-free DNA machine for the sensitive detection of ATP. According to your nice suggestions, we have listed a table to compare the analytical performance (LOD, linear range, etc.) with other papers, as shown in Table 3.
Another aspect concerns the English language presentation. For me, it will be require a mother tongue revision, especially for the sentence constructions, idiomatic expression and a more synthetic style, get to the point expeilly for conclusions, extracted from results and experimental data.
Thanks for your nice suggestions. This manuscript has been checked by a native English speaker. We believe that the manuscript has greatly benefit from this revising.
Reviewer 2 Report
The manuscript entitled "A Label-Free Fluorescent DNA Machine for Sensitive Cyclic Amplification Detection of ATP" report on the development of DNA machine based on the combination of several logic gates allowing to obtain a sensitive ATP detection through an effective recycling instead of using yes/no response. The manuscript is well developed in all section therefore I would suggest to accept as it is.
Author Response
Thank you for your positive comments.
Reviewer 3 Report
The paper is well written. On the other hand, some modifications are needed as follows:
Section 1. “Introduction”
The introduction is clear and all the objectives well stated. Some recent technology developments are missing such as:
Metamaterials [From metamaterials to metadevices, Nature Materials 11, 917–924, 2012]
Near-zero-index materials [Near-zero-index wires, Optics express 25 (20), 23699-23708, 2017]
It would be beneficial for the reader if authors include such technologies in the introduction section to have a complete picture of the state-of-art.
Section 3. “Discussion and Analysis”
1) The structure is well explained. The used model is interesting, but a more detailed discussion it is necessary. Consider in your work the following techniques: multi-layer structures, Transmission-Line-Theory, Composite media and Scattering.
Explain in detail what are the advantages/disadvantages and similarities/differences of your model compared to the ones above-mentioned.
2) To explore the device behavior, authors should consider the following interesting phenomena: electric/magnetic currents and surface waves.
How they can affect the device performance response?
3) The paper lack in applications examples. Take into consideration the following: telecommunications, sensing & diagnostics, antennas, measurements and automotive.
I would suggest explaining how you can use your device in such applications (please highlight what's new in yours).
Section 4. “Conclusion”
1) No limitations of the proposed method have been highlighted.
2) No future improvements/works have been discussed.
Author Response
To Referee: 3
The paper is well written. On the other hand, some modifications are needed as follows:
Section 1. “Introduction”
The introduction is clear and all the objectives well stated. Some recent technology developments are missing such as:
Metamaterials [From metamaterials to metadevices, Nature Materials 11, 917–924, 2012]
Near-zero-index materials [Near-zero-index wires, Optics express 25 (20), 23699-23708, 2017]
It would be beneficial for the reader if authors include such technologies in the introduction section to have a complete picture of the state-of-art.
Thank you very much for your suggestion. We have discussed the artificial electromagnetic metamaterials and near-zero-index materials for improving the sensitivity of biosensors in the introduction section. And these relative references have also been cited.
Section 3. “Discussion and Analysis”
1) The structure is well explained. The used model is interesting, but a more detailed discussion it is necessary. Consider in your work the following techniques: multi-layer structures, multi-layer structures, Composite media and Scattering.
Explain in detail what are the advantages/disadvantages and similarities/differences of your model compared to the ones above-mentioned.
Thank you for your valuable suggestions to improve the quality of our manuscript. The probe of fluorescence DNA biosensor was a liquid including DNA-T, DNA-M, DNA-M', GO and PicoGreen dye, not solid electrode material. So, we are very sorry. We couldn't consider multi-layer structures, multi-layer structures etc.
In addition, in the conventional analytical systems, it is difficult to achieve the simultaneous signal amplification, background signal suppression, label-free fluorescence, enzyme-free ATP detection. In this work, target ATP cyclic amplification, GO suppression background signals, and PicoGreen dye were used to construct a label-free fluorescent and enzyme-free DNA machine for the sensitive detection of ATP. According to your nice suggestions, we have listed a table to compare the analytical performance (LOD, linear range, etc.) with other papers, as shown in Table 3 of the manuscript.
2) To explore the device behavior, authors should consider the following interesting phenomena: electric/magnetic currents and surface waves.
How they can affect the device performance response?
Thanks for your nice suggestions. Our paper studied fluorescence DNA biosensors, not physical sensors. The probe of fluorescence DNA biosensor was a liquid including DNA-T, DNA-M, DNA-M', GO and PicoGreen dye, not solid electrode material. So, we are very sorry. We couldn't consider electric/magnetic currents and surface waves etc.
3) The paper lack in applications examples. Take into consideration the following: telecommunications, sensing & diagnostics, antennas, measurements and automotive.
I would suggest explaining how you can use your device in such applications (please highlight what's new in yours).
Thank you for your valuable suggestions to improve the quality of our manuscript. The DNA machine has been used to detect ATP concentration in real urine samples in the 96th row of the manuscript (See the title: 3.5. Using the DNA machine for ATP concentration detection in real urine samples). Table 4 in manuscript shows that the DNA machine exhibits good recoveries in the range of 95.76–102.03%, indicating that the proposed DNA machine has great advantages for ATP detection in clinical diagnosis.
In addition, In the conventional analytical systems, it is difficult to achieve the simultaneous signal amplification, background signal suppression, label-free fluorescence, enzyme-free ATP detection. In this work, target ATP cyclic amplification, GO suppression background signals, and PicoGreen dye were used to construct a label-free fluorescent and enzyme-free DNA machine for the sensitive detection of ATP.
Section 4. “Conclusion”
1) No limitations of the proposed method have been highlighted.
Thank you for your valuable suggestions to improve the quality of our manuscript. The detection range of the proposed method was found to be from 100 to 600 nM (R2 = 0.99108) with a LOD of 127.9 pM, which has written in the 206th line of the manuscript.
2) No future improvements/works have been discussed.
Thanks for your nice suggestions. In the future, the sensitivity of the DNA machine for ATP detection can be improved by catalyzing hairpin assembly, which has written in the 211th line of the manuscript.
Again, many thanks for your editorial endeavors on behalf of all of the authors and we do appreciate very much the constructive comments and good suggestions from the reviewers such that we were able to improve the overall quality and clarity of our paper. Hopefully, we could have our article been considered of publication in your journal. We really believe this work would be sufficient novelty and impact to appeal to your readership. Should there been any other corrections we could make, please feel free to contact us.
Sincerely Yours,
Jie Du
College of Materials and Chemistry Engineering,
Hainan University,
Haikou 570228, PR China.
E-mail: dujie@hainu.edu.cn